# Creatine kinase and its muscle/brain isoenzyme as potential biomarkers for cardiovascular disease risk: A cross-sectional analysis of a population-based cohort

Mahla Izadpanah[1], Elahe Pourahmadi[2], Hamideh Safarian Bana[3], Elham Dehghani[4], Masoumeh Akbari[5], Fatemeh Asgharian Rezae[6], Fatemeh Yarmohammadi[7], Ashraf Alnattah[5], Saeid Eslami[5], Zhila Taherzadeh[8,9]*

1 Department of Pharmacodynamy and Toxicology, School of Pharmacy, Mashhad University of Medical Sciences, Mashhad, Iran, 2 Social Departments of Health Research Center, Mashhad University of Medical Sciences, Mashhad, Iran, 3 Metabolic Syndrome Research Center, Faculty of Medicine, Mashhad University of Medical Sciences, Mashhad, Iran, 4 Faculty of Medicine, Shahid Sadoghi University of Medical Science, Yazd, Iran, 5 Medical Informatics Department, Faculty of Medicine, Mashhad University of Medical Sciences, Mashhad, Iran, 6 Student Research Committee, Faculty of Pharmacy, Mashhad University of Medical Sciences, Mashhad, Iran, 7 Medical Biology Research Center, Health Technology Institute, Kermanshah University of Medical Sciences, Kermanshah, Iran, 8 Applied Biomedical Research Center, Mashhad University of Medical Sciences, Mashhad, Iran, 9 Targeted Drug Delivery Research Center, Mashhad University of Medical Sciences, Mashhad, Iran

* Taherzadehzh@mums.ac.ir

## Abstract

### Introduction

This study examines the correlation between creatine kinase (CK) and its muscle/brain isoenzyme (CK-MB) in relation to cardiovascular risk factors, highlighting sex disparities.

### Methods

A three-year cross-sectional study was conducted among 246 adults recruited from the Mashhad University of Medical Sciences (MUMS) Persian Cohort. CK and CK-MB concentrations were measured using standardized laboratory assays. Correlations and multivariable linear models were used to assess associations with cardiovascular risk indicators.

### Results

Men exhibited significantly higher log-transformed CK and CK-MB levels than women. Both enzymes showed weak-to-moderate correlations with selected cardiovascular parameters. Log CK was modestly associated with diastolic blood pressure (DBP), creatinine, and PWV ($r < 0.29$), whereas log CK-MB showed positive associations with calcium, potassium, DBP, and PWV. Many univariable associations

**Data availability statement:** The datasets generated and analyzed during the current study are not publicly available due to ethical restrictions and privacy concerns regarding human participant data, as mandated by the Institutional Review Board (IRB) of Mashhad University of Medical Sciences. However, de-identified data can be made available to qualified researchers upon reasonable request, subject to approval by the MUMS IRB. Data access inquiries should be directed to the MUMS Ethics Committee at ethics@mums.ac.ir or cohort@mums.ac.ir.

**Funding:** The author(s) received no specific funding for this work.

**Competing interests:** The authors have declared that no competing interests exist.

**Abbreviations:** ADP, Adenosine diphosphate; ATP, Adenosine triphosphate; AIP, Atherogenic Index of Plasma; Aix, Augmentation Index; BMI, Body Mass Index; BP, Blood Pressure; BUN, Blood Urea Nitrogen; $Ca^{2+}$, Calcium ion; CK, Creatine Kinase; CK-MB, Creatine kinase-muscle/brain; $Cl^-$, Chloride ion; CVD, Cardiovascular diseases; CVD-H, Cardiovascular Events History; DBP, Diastolic Blood Pressure; EDTA, Ethylenediaminetetraacetic Acid; FBS, Fasting Blood Sugar; $H^+$, Hydrogen ion; HDL, High-density lipoprotein; $K^+$, Potassium ion; Log, Logarithm; MPWV, Mean Pulse Wave Velocity; MUMS, Mashhad University of Medical Sciences; $Mg^{2+}$, Magnesium ion; $Na^+$, Sodium ion; $PO_4^{3-}$, Phosphate ion; PWV, Pulse Wave Velocity; SBP, Systolic Blood Pressure; SD, Standard Deviation; TC, Total Cholesterol.

attenuated after adjustment, with only serum calcium remaining independently related to log CK-MB (p = 0.03).

## Conclusion

In this cross-sectional cohort, log CK and log CK-MB showed weak but statistically detectable associations with selected cardiovascular measures. These findings are hypothesis-generating and should be interpreted cautiously; they do not support clinical recommendations without prospective validation in larger studies.

## Introduction

Cardiovascular disease (CVD) remains the leading cause of death worldwide, motivating the search for accessible indicators that can refine risk assessment. Creatine kinase (CK) is essential to cellular energy transfer, regenerating ATP via the phosphocreatine shuttle and supporting ATP-dependent processes central to cardiovascular physiology (e.g., vascular smooth-muscle contraction and ion transport) [1,2]. Observational studies have indicated correlations between elevated circulating CK levels and blood pressure measurements, exhibiting variation based on sex and anthropometric characteristics, which likely signifies differences in skeletal muscle mass [3,4]. These observations indicate CK may reflect elements of systemic bioenergetics pertinent to cardiovascular phenotypes and highlight the necessity of considering demographic and body composition factors in interpreting CK levels [5,6].

Because ATP-driven membrane transport (e.g., $Na^+/K^+$-ATPase and $Ca^{2+}$ handling) underpins cardiovascular function, variation in energy supply could plausibly relate to blood-pressure regulation and other risk markers [7–9]. Prior findings are inconsistent and largely cross-sectional; causal inference is not established, and effect sizes are typically small [10].

Aside from total CK, the muscle/brain isoenzyme CK-MB requires specific consideration. While CK primarily indicates skeletal muscle involvement, CK-MB is comparatively cardiac-specific and can reflect minor myocardial workload or stress. CK-MB is established for diagnosing acute myocardial ischemia [11,12], yet its broader epidemiologic relevance in community cohorts remains less well characterized [13].

Our approach in this study was to evaluate CK and CK-MB simultaneously, allowing a side-by-side comparison of two widely available biomarkers previously linked to CVD. This joint analysis provides a concise framework to consider complementary information from CK (systemic, muscle-influenced bioenergetics) and CK-MB (more cardiac-specific signal). In doing so, we revisit established patterns and offer a focused perspective on how CK-MB may relate—albeit subtly in community settings—to the network of cardiovascular risk markers.

We examined whether circulating CK and CK-MB are associated with cardiovascular risk factors in a community sample, with particular attention to sex differences and body-composition context. Our aim was to assess the population-level relevance of these readily available laboratory measures and to generate hypotheses for future

prospective studies. We analyzed CK and CK-MB on the log scale (to address right-skew), emphasized effect sizes over statistical significance, and interpreted any observed associations as exploratory rather than diagnostic or prognostic.

## 2. Materials and methods

### 2.1. Study design and participants

This was a cross-sectional observational analysis derived from the MUMS Persian Cohort. We examined the association between serum CK and CK-MB and cardiovascular risk factors among individuals who attended the cohort health-monitoring center between September 2019 and October 2021 and met pre-specified eligibility criteria. Participants were selected from routine check-ups and were included when complete data were available for blood pressure, cardiovascular disease history (CVD-H), medications, and physical activity. All data were collected at a single time point per participant.

Written informed consent had been obtained at the time of cohort enrollment using Institutional Review Board–approved forms; the protocol was approved by the Ethical Committee of Mashhad University of Medical Sciences (IR. MUMS.REC.1396.414). As an exploratory observational study with no intervention or hypothesis testing, no formal sample-size calculation was performed and trial registration (e.g., ClinicalTrials.gov) was not required. The aim was to describe associations between CK/CK-MB and cardiovascular risk factors in a population-based framework.

### 2.2. Inclusion and exclusion criteria

Participants were identified from the cohort database based on data completeness and clinical relevance.

Inclusion criteria. Adults with available measurements of serum CK and CK-MB, blood pressure, cardiovascular history, and relevant biochemical and anthropometric variables.

Exclusion criteria. To minimize confounding from conditions that alter CK or CK-MB, we excluded individuals with disorders affecting skeletal or cardiac muscle integrity or metabolism (e.g., alcohol-related liver disease, thyrotoxicosis, Cushing's syndrome, connective-tissue disease, active infection, rheumatoid arthritis, dermatomyositis, polymyositis). Pregnant individuals and those with BMI < 18.5 kg/m² (used as a surrogate for low muscle mass, per prior work [3] were excluded. We also excluded persons using medications known to substantially influence CK/CK-MB (e.g., captopril, colchicine, alcohol misuse, lovastatin, lithium, lidocaine, propranolol).

### 2.3. Data collection and measurements

Trained interviewers obtained demographic and clinical information using standardized Ministry of Health questionnaires (lifestyle, nutrition, and cardiovascular history). Anthropometries (weight, height, waist and hip circumference) were measured to calculate BMI.

Participants provided fasting blood (≥5 hours fast; ≥ 14 hours caffeine abstinence) and spot urine samples. Laboratory analyses included electrolytes (sodium, potassium, calcium, phosphate), fasting blood sugar, blood urea nitrogen, creatinine, and lipid measures (total cholesterol, HDL, triglycerides), using a BT1500 auto-analyzer. The levels of CK and CK-MB in serum were measured using a turbidimetric immunoassay applied on a fully automated Hitachi analyzer, following the outlined procedure provided by the manufacturer.The clinical reference intervals used to determine normal ranges were sourced directly from the manufacturer's package inserts for the individual reagent kits used in the BT1500 auto-analyzer (Biosystems S.A., Barcelona, Spain). The established CK reference ranges are 24–170 U/L for women, 24–195 U/L for men, and a maximum of 24 U/L for CK-MB.

All assays incorporated internal quality controls and were performed in duplicate; the mean of duplicates was analyzed. Lipid-related risk was summarized by the Atherogenic Index of Plasma AIP = log10[TG (mmol/L)/HDL-cholesterol (mmol/L)], which served as the primary lipid metric for analyses.

Physical activity was assessed using a questionnaire validated in European studies and subsequently evaluated for validity and reliability by Iranian researchers. Daily exercise hours were recorded, and activity was categorized as sedentary, mild, moderate, or vigorous (including sitting and low-intensity activity).

All records were de-identified at collection using unique IDs; investigators analyzed anonymized data.

### 2.4. Cardiovascular evaluation

Blood pressure was measured in duplicate after seated rest using an analog sphygmomanometer with appropriately sized cuffs. Pulse wave velocity (PWV), a marker of arterial stiffness, was measured with a SphygmoCor XCEL device. Hypertension status was coded as treated-hypertensive (current antihypertensive therapy) versus normotensive at assessment. CVD-H was defined a priori as ≥1 prior physician-diagnosed myocardial infarction, angina, or stroke, coded as binary (0 = none; 1 = ≥1 event). CVD-H was assessed within a GLM structure for exploratory comparison.

### 2.5. Statistical analysis

All analyses were performed using SPSS version 26. To account for the right skew and stabilize the variance, serum CK and CK-MB data were log transformed. Continuous variables are expressed as mean ± SD, whereas categorical data are portrayed as frequencies or percentages. We investigated the relationships between log CK or log CK-MB and cardiovascular risk variables using both univariable (Pearson's or point-biserial) and multivariable general linear models (GLM) with sex and hypertension status. The model's assumptions of normality, linearity, and homoscedasticity were confirmed and found to be acceptable. We focused on effect sizes (β, r) with 95% confidence intervals and included a descriptive explanation for p-values (α = 0.05). A post-hoc power analysis indicated >80% power to detect correlations of r ≥ 0.25 at α = 0.05.

We classified correlation strength as strong if r > 0.80, moderate if between 0.60 and 0.79, and weak when r < 0.29. We did not correct for multiple comparisons, as this research was focused on exploration and generating hypotheses.

## 3. Results

### 3.1. Clinical and Biochemical Characteristics

Table 1 summarizes the clinical and biochemical characteristics of the study population. Of the participants, 103 were men (41.9%) and 143 were women (58.1%); 43 (17.5%) were treated-hypertensive and 203 (82.5%) were normotensive at

**Table 1. The clinical characteristics of participants.**

| Variables | Minimum | Maximum | Mean | SD |
|---|---|---|---|---|
| Age (years) | 35.00 | 74.00 | 48.05 | 10.87 |
| BMI (kg/m²) | 15 | 44 | 27.26 | 4.24 |
| Calcium (mg/dL) | 8.60 | 10.40 | 9.45 | 0.39 |
| Phosphate (mg/dL) | 3.00 | 5.00 | 3.69 | 0.50 |
| Sodium (mmol/L) | 38.11 | 142.30 | 138.38 | 6.55 |
| Potassium (mmol/L) | 3.10 | 5.10 | 4.33 | 0.28 |
| Creatinine (mg/dL) | 0.65 | 1.44 | 1.00 | 0.16 |
| CK (U/L) | 34.00 | 548.00 | 100.67 | 59.18 |
| CK-MB (U/L) | 3.50 | 21.2 | 7.21 | 2.80 |
| PWV (m/s) | 2.14 | 10.30 | 6.43 | 1.40 |
| AIP (log[TG/HDL-C]) | −0.6 | 0.6 | −0.2 | 0.23 |

**Legend:** Abbreviations: CK: creatine kinase; CK-MB: creatine kinase-muscle/brain; SBP: systolic blood pressure; DBP: diastolic blood pressure; BMI: body mass index; AIP: Atherogenic Index of Plasma.

assessment (Table 2). Participants were predominantly middle-aged (mean age 48.05 ± 10.87 years), a group at increased risk for cardiovascular disease.

Electrolytes (calcium, phosphate, sodium, potassium) were within expected physiological ranges, and the renal indices (creatinine and BUN) showed that kidney function was still good, which meant that renal indices were within expected ranges at the group level (Table 1). Overall BMI fell in the overweight range (mean 27.26 ± 4.24 kg/m²), with no statistically significant difference.

AIP averaged −0.20 ± 0.23 (range −0.60 to 0.60), indicating predominantly low atherogenic risk with a small high-risk tail (>0.21). The pulse wave velocity distribution suggested moderate arterial stiffness (mean ≈ 6.4 m/s).

The main variables of interest, CK and its isoenzyme CK-MB, had average values of 100.67 U/L and 7.21 U/L, respectively. The large standard deviations for both CK (59.18) and CK-MB (2.80) showed that there is a lot of variation between people, which suggests that some people may have higher levels. On the original scale, central tendencies were approximately 101 U/L for CK and 7.2 U/L for CK-MB (Table 1).

### 3.2. Sex-Specific CK/ CK-MB distribution

As illustrated in Figs 1 and 2, significant sex differences were observed in both CK (Fig 1) and CK-MB (Fig 2). Log-transformed CK and CK-MB levels were consistently higher in males compared to females (Figs 1 and 2). These are descriptive differences consistent with enzyme biology and body-composition context. Effect sizes and distributional

**Table 2. Correlation Coefficients of log CK with selected cardiovascular risk factors.**

| Status | Gender Status Count (Percentage) | | Total | Hypertension Status Count (Percentage) | |
|---|---|---|---|---|---|
| Variables | Male 103 (%41.9) Sig. (1-tailed) Correlation Coefficient(r) | Female 143 (%58.1) Sig. (1-tailed) Correlation Coefficient(r) | Total 246 (%100) Sig. (1-tailed) Correlation Coefficient(r) | Without Hypertension 203 (%82.5) Sig. (1-tailed) Correlation Coefficient(r) | With Hypertension 43 (%17.5) Sig. (1-tailed) Correlation Coefficient(r) |
| Gender | -------- | --------- | −0.27 (0.001) | 0.001 (−0.29) | 0.09 (−0.20) |
| Age | 0.26 (−0.11) | 0.3 (0.04) | 0.48 (0.002) | 0.25 (−0.04) | 0.1 (0.19) |
| BMI | 0.09 (0.17) | 0.06 (0.25) | 0.04 (0.25) | 0.07 (0.17) | −0.15 (0.17) |
| Ca | **0.01* (0.23[a])** | **0.24 (−0.06)** | **0.06 (0.1)** | 0.04 (0.13) | 0.42 (0.03) |
| Na | 0.44 (−0.01) | 0.21 (−0.06) | 0.14 (−0.06) | 0.27 (−0.04) | 0.41 (0.03) |
| K | **0.28 (0.05)** | **0.4 (0.02)** | **0.02* (0.12[a])** | **0.002* (0.22 [a])** | **0.2 (−0.13)** |
| Ph | 0.27 (0.06) | **0.01* (0.26[a])** | **0.06 (0.1)** | 0.06 (0.11) | **0.39 (0.04)** |
| FBS | 0.33 (−0.04) | 0.38 (0.02) | 0.22 (0.05) | 0.15 (−0.07) | 0.14 (−0.17) |
| BUN | 0.13 (0.11) | 0.39 (−0.02) | 0.35 (0.02) | 0.3 (0.03) | 0.18 (−0.14) |
| Creatinine | **0.04* (0.19[a])** | 0.36 (0.03) | **0.001* (0.23[a])** | **0.001* (0.3 [a])** | 0.41 (−0.03) |
| CHOL | 0.25 (0.06) | 0.48 (0.003) | 0.46 (0.006) | 0.32 (0.03) | 0.3 (0.08) |
| AIP | −0.056 (0.59) | −0.057 (0.51) | 0.96 (0.003) | 0.14 (0.07) | 0.003 (−0.42) |
| SBP | 0.43 (0.02) | 0.17 (0.08) | 0.21 (−0.05) | 0.06 (0.12) | 0.46 (−0.01) |
| DBP | **0.01* (0.23[a])** | 0.05 (0.14) | **0.01* (0.15[a])** | **0.002* (0.24[a])** | 0.31 (0.08) |
| PWV | 0.4 (0.02) | 0.34 (0.04) | **0.001* (0.27[a])** | 0.17 (0.08) | 0.15 (−0.19) |
| CVD-H | **0.13 (−0.11)** | **0.34 (0.03)** | 0.07 (0.11) | **0.001* (−0.29 [a])** | 0.17 (−0.14) |

**Legend:** Correlation coefficients (r) between log CK and cardiovascular risk factors. The direction of association is given by the sign of r.

*Correlation is significant at the 0.05 level (1-tailed) between the two variables

[a]Weak positive (negative) linear relationship

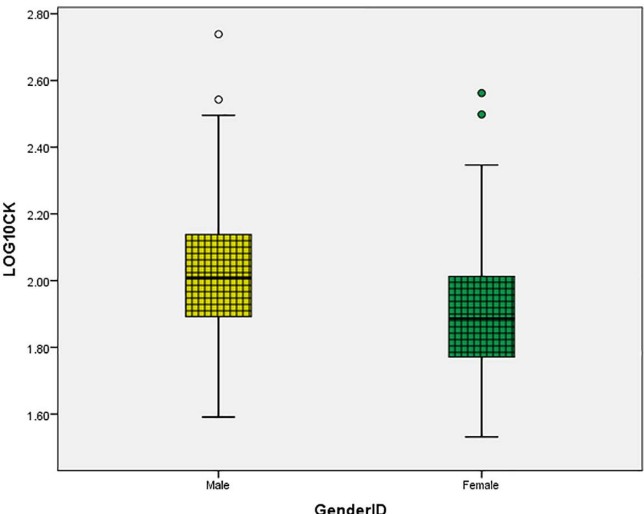

**Fig 1. Sex-specific CK distribution.** Distribution of log-transformed CK (levels by sex.

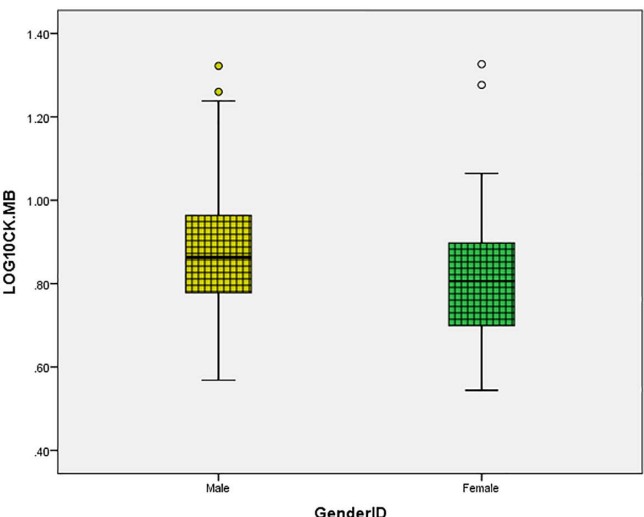

**Fig 2. Sex-specific CK-MB distribution.** Distribution of log-transformed CK-MB levels by sex.

summaries appear in Figs 1 and 2, and the corresponding table notes. The mean CK levels were 2.01±0.2 in men and 1.9±0.18 in women (Fig 1). Similarly, the mean CK-MB levels were 0.87±0.14 in men and 0.8±0.13 in women (Fig 2).

### 3.3. CK/CK-MB in Normotensive versus Hypertensive Groups

Log CK and log CK-MB distributions were similar between normotensive and treated-hypertensive participants at the time of assessment (Figs 3 and 4 respectively), with a substantial overlap of medians and interquartile ranges. Among normotensive individuals, the mean CK was 1.9±0.2, whereas it was 1.9±0.19 in treated-hypertension (Fig 3). In hypertensive individuals, the mean CK-MB was 0.87±0.11, compared to 0.82±0.14 in normotensive participants(Fig 4).

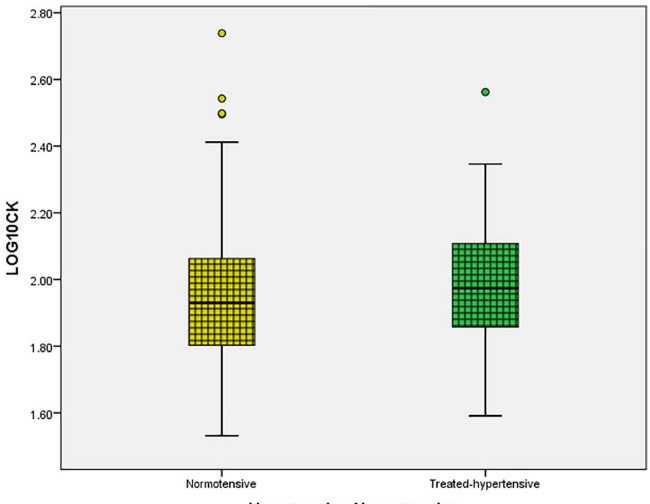

**Fig 3. Distribution of CK in normotensive versus treated-hypertensive individuals.** Comparison of log-transformed CK levels between normotensive participants and those with treated-hypertension.

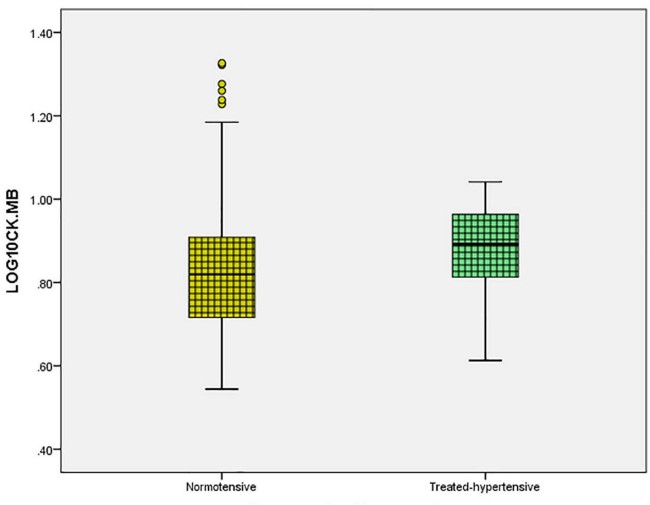

**Fig 4. Distribution of CK-MB in normotensive versus treated-hypertensive individuals.** Comparison of log-transformed CK-MB levels between normotensive participants and those with treated-hypertension.

### 3.4. Associations of CK with cardiovascular risk factors

As shown in Table 2, in the total sample, higher log CK correlated positively with creatinine (r = 0.23, p = 0.001), DBP (r = 0.15, p = 0.01), serum potassium (K) (r = 0.12, p = 0.02), and mean PWV (r = 0.27, p = 0.001), and differed significantly by gender (r = −0.27, p = 0.001) (Table 2).

In general, the effect sizes were small to moderate (r ≈ 0.12–0.30). The most significant associations were between PWV (r ≈ 0.27) and AIP among treated hypertensives (r ≈ −0.42). The directionality suggests that higher CK is usually linked to small increases in DBP, K, PWV, and creatinine. This shows that muscle-derived enzymes, electrolyte handling, and vascular function are all linked (Table 2).

In univariable regression models (Table 3), log CK was significantly associated with DBP (β = 0.17, p = 0.01) and CVD-H (β = 0.14, p = 0.04) but not with other variables. After adjustment in multivariable analyses, these associations attenuated and were no longer statistically significant, indicating that the crude effects largely reflect shared variance with sex, body composition, or other covariates.

Taken together, these findings support the primary hypothesis that log CK is associated with several cardiovascular risk markers, although the relationships are weak and not independent after adjustment. The data imply that observed CK–blood-pressure links likely represent secondary effects of physiological factors such as muscle mass, renal function, and electrolyte balance rather than direct causal pathways.

### 3.5. Associations of CK-MB with cardiovascular risk factors

As shown in Table 4, in the entire population, elevated log CK-MB exhibited positive correlations with calcium (r = 0.29, p = 0.001), potassium (r = 0.30, p = 0.001), SBP (r = 0.22, p = 0.001), DBP (r = 0.23, p = 0.001), PWV (r = 0.18, p = 0.01), and creatinine (r = 0.28, p = 0.001). A smaller positive correlation was noted with CVD-H (r = 0.12, p = 0.003). As expected, gender exhibited an inverse correlation (r = −0.25, p = 0.001), aligning with elevated CK-MB levels in males (Table 4).

Overall, these results support the idea that log CK-MB is linked to several cardiovascular and metabolic markers, but the links are mostly weak and only show correlation.

In univariable linear regression analyses (Table 5), log CK-MB was significantly associated with gender (β = −0.07, p = 0.001), Ca (β = 0.31, p < 0.001), K (β = 0.12, p = 0.001), SBP (β = 0.19, p = 0.001), DBP (β = 0.22, p = 0.005), and PWV (β = 0.20, p = 0.01). BUN was borderline significant (β = 3.5, p = 0.05). In the multivariable general linear model (GLM), only serum calcium remained statistically significant (Mean Square = 0.19, F = 1.52, p = 0.03). Other variables—including Na, K, DBP, and PWV—lost significance (p > 0.05), indicating attenuation because of collinearity or confounding.

Collectively, the data identify serum calcium as the most consistent independent correlate of log CK-MB after adjustment. The blood-pressure and arterial-stiffness associations observed in univariable screening were not robust in multivariable analysis. These patterns suggest that CK-MB–vascular relationships are largely secondary to shared physiological pathways (electrolyte balance, muscle metabolism, renal function) rather than direct effects. While most associations were weak, their biological plausibility and internal consistency warrant further study in larger and better-powered samples.

**Table 3. Univariable log CK with selected cardiovascular risk factors.**

| Dependent Variables | Univariable Regression | | |
|---|---|---|---|
| | Coefficient(β) | 95% CI | P-value |
| Gender | −0.1 | 0.057 - 0.158 | 0.001** |
| Ca | −0.09 | 0.052 - 0.139 | 0.18 |
| K | 0.008 | 0.021 - 0.154 | 0.9 |
| Ph | 0.01 | −0.009 - 0.059 | 0.85 |
| Creatinine | 0.04 | −0.018 - 0.184 | 0.49 |
| SBP | 0.08 | −0.003 - 0.007 | 0.24 |
| DBP | 0.17 | −0.009 - 0.012 | 0.01** |
| CVD-H | 0.14 | −0.005 - 0.113 | 0.04** |

**Legend:** Univariable linear regression (β) of log CK on cardiovascular risk factors. Coefficients indicate direction and estimated size in unadjusted models.

Independent Variable: log CK

** The independent variable is significantly associated with the dependent variable

**Table 4. Correlation Coefficients of log CK-MB and selected cardiovascular risk factors.**

| Status | Gender Status Count (Percentage) | | Total | Hypertension Status Count (Percentage) | |
|---|---|---|---|---|---|
| Variables | Male 103 (%41.9) Sig. (1-tailed) Correlation Coefficient(r) | Female 143 (%58.1) Sig. (1-tailed) Correlation Coefficient(r) | 246 (%100) Sig. (1-tailed) Correlation Coefficient(r) | Without Hypertension 203 (%82.5) Sig. (1-tailed) Correlation Coefficient(r) | With Hypertension 43 (%17.5) Sig. (1-tailed) Correlation Coefficient(r) |
| Gender | ------ | ------ | −0.250.001 | 0.001 (−0.25) | 0.14 (−0.16) |
| Age | 0.077 | 0.066 | 0.06 | 0.25 (0.05) | 0.36 (0.05) |
| BMI | 0.12 (0.11) | 0.13 (0.06) | 0.1 (0.06) | 0.07 (0.14) | 0.12 (0.21) |
| Ca | 0.000* (0.337) | 0.005* (0.22[a]) | 0.001* (0.29[a]) | 0.001* (0.33[a]) | 0.16 (0.15) |
| Na | 0.135 | 0.001* (−0.27[a]) | 0.001* (−0.22[a]) | 0.33 (−0.03) | 0.46 (−0.01) |
| K | 0.13 (0.3) | 0.001* (0.3[a]) | 0.001* (0.3[a]) | 0.001* (0.26[a]) | 0.13 (0.17) |
| Ph | 0.4 | 0.003* (0.24[a]) | 0.07 | 0.12 (0.08) | 0.06 (0.35[a]) |
| FBS | 0.17 | 0.4 (0.15) | 0.003* (0.18) | 0.33 (−0.03) | 0.44 (−0.02) |
| BUN | 0.39 | 0.08 | 0.03* (0.12[a]) | 0.02* (0.14 [a]) | 0.39 (−0.08) |
| Creatinine | 0.001* (0.32[b]) | 0.06 | 0.001* (0.28[a]) | 0.001* (0.32[a]) | 0.32 (0.07) |
| CHOL | 0.44 | 0.01* (0.19[a]) | 0.14 | 0.01* (0.16 [a]) | 0.07 (−0.23) |
| AIP | −0.06 (0.55) | 0.03 (0.72) | 0.04 (0.48) | 0.07 (0.16) | 0.2 (−0.3) |
| SBP | 0.02* (0.2[a]) | 0.01* (0.1[a]) | 0.001* (0.22[a]) | 0.004 (0.20) | 0.39 (−0.04) |
| DBP | 0.002* (0.3[a]) | 0.05 | 0.005* (0.5) | 0.001* (0.23[a]) | 0.36 (0.05) |
| PWV | 0.07 | 0.03* (0.19[a]) | 0.01* (0.2[a]) | 0.01* (0.18 [a]) | 0.48 (0.007) |
| CVD-H | 0.04* (0.32[b]) | 0.04* (0.15[a]) | 0.003* (0.12[a]) | 0.001 (−0.25) | 0.14 (−0.16) |

**Legend:** Multivariable general linear model (β) for log CK-MB.

*Correlation is significant at the 0.05 level (1-tailed) between the two variables

[a]Weak positive (negative) linear relationship

### 3.6. Subgroup analyses (sex and hypertension status)

In sex-stratified exploratory summaries (Tables 2 and 4), among men, log CK correlated with calcium (Ca) (r = 0.23, p = 0.01), creatinine (r = 0.19, p = 0.04), and DBP (r = 0.23, p = 0.01). Among women, only phosphate (Ph) showed a positive correlation (r = 0.26, p = 0.01). In normotensive participant, log CK was positively linked to potassium (r = 0.22, p = 0.002) and DBP (r = 0.24, p = 0.002), but negatively linked to CVD-H (r = −0.29, p = 0.001). In the treated hypertensive subgroup, AIP exhibited a moderate inverse correlation (r = −0.42, p = 0.003).

Men exhibited more robust correlations with Ca (r = 0.34, p < 0.001), creatinine (r = 0.32, p = 0.001), DBP (r = 0.30, p = 0.002), SBP (r = 0.20, p = 0.02), and a trend with K (r = 0.30, p = 0.13). In women, log CK-MB showed positive associations with Ca (r = 0.22, p = 0.005), K (r = 0.30, p = 0.001), phosphate (Ph) (r = 0.24, p = 0.003), total cholesterol (CHOL) (r = 0.19, p = 0.01), PWV (r = 0.19, p = 0.03), and AIP (r = 0.72, p = 0.03). The latter represents an unusually strong lipid–enzyme correlation and may reflect a small subgroup with highly atherogenic lipid profiles.

In normotensive participants, positive correlations were again evident for Ca, K, SBP, DBP, PWV, and creatinine, while CVD-H correlated negatively (r = −0.25, p = 0.001). Among treated-hypertensive participants, most associations attenuated and became non-significant, suggesting that treatment status or medication effects may obscure the enzyme–risk relationships.

Effect magnitudes were predominantly small to moderate (r ≈ 0.18–0.33), except for a few larger sex- or lipid-specific findings (e.g., AIP in women).

**Table 5. Univariable and General Linear Model of log CK-MB with selected cardiovascular risk factors.**

| Dependent Variables | Univariable Regression | | | General Linear Model | | |
|---|---|---|---|---|---|---|
| | Coefficient (β) | 95% CI | P-value | Mean Square | F | P-value |
| Gender | −0.07 | 0.052-0.139 | 0.001** | ------- | ------ | ------- |
| Ca | 0.31 | 0.052-0.139 | 0.000** | 0.19 | 1.52 | 0.03 ** |
| Na | −0.03 | −0.001-0.004 | 0.63 | 47 | 0.56 | 0.98 |
| K | 0.12 | 0.021-0.154 | 0.001** | 0.08 | 0.9 | 0.54 |
| FBS | −0.03 | −0.001-0.000 | 0.56 | ------- | ------ | ------- |
| BUN | 3.5 | −0.009- 0.059 | 0.05 | ------- | ------ | ------- |
| Creatinine | 0.04 | −0.018-0.184 | 0.52 | ------- | ------ | ------- |
| SBP | 0.19 | −0.003-0.007 | 0.001** | ------- | ------ | ------- |
| DBP | 0.22 | −0.009-0.012 | 0.005** | 77 | 0.79 | 0.82 |
| PWV | 0.2 | −0.010-0.029 | 0.01** | 2.39 | 1.33 | 0.11 |
| CVD-H | 0.12 | −0.005-0.113 | 0.07 | ------- | ------ | ------- |

**Legend:** Independent Variable: CK-MB

** The independent variable is significantly associated with the dependent variable

## 4. Discussion

In this community cohort, men showed higher log CK and log CK-MB distributions than women, although normotensive and treated-hypertensive participants had essentially identical distributions. After multivariable model correction, log CK had no independent relationship with blood pressure or arterial stiffness data. In contrast, for log CK-MB, serum calcium maintained a modest but independent relationship, albeit crude correlations with SBP/DBP and PWV were reduced in adjusted analysis. The observed associations exhibited a moderate magnitude and were largely correlative.

### 4.1. Clinical relevance of CK and CK-MB Levels

The observed sex-specific variations, with elevated CK and CK-MB levels in males, align with the enzymatic properties and disparities in body composition, such as increased lean muscle mass in men [14]. The higher CK levels in men could be attributed to physiological factors such as testosterone, which can raise CK levels and influence skeletal muscle metabolism [15]. Additionally, the research determined that CK-MB's specific role extends beyond its existence, and this reflects underlying physiological processes influenced by sex [14]. According to existing research, CK-MB is still a reliable marker for determining cardiovascular risk and assessing acute ischemia [16]. However, it is recommended to re-evaluate it because of its unique insights concerning sex dynamics and their effects on cardiovascular health [17,18].

### 4.2. Blood pressure considerations

The association between CK and hypertension has been confirmed in several studies in Western societies [5,19–21]. Many mechanisms have been proposed for CK's participation in hypertension. The energy regulator enzyme CK catalyzes the reversible transfer of a high-energy phosphate group between phosphocreatine and ADP, producing ATP. CK provides cardiovascular contractile proteins with ATP for contraction. Thus, elevated CK activity may increase contractile protein ADP/ATP balance, contractility, and myosin ATPase activity [20,21]. CK is also tightly linked to energy regulator enzymes such as Na+/K+, Ca2+-ATPase, myosin ATPase, and myosin light chain kinase. Thus, CK activity affects enzyme-regulated activities including salt retention and cardiovascular contractility [22]. The vascular smooth muscles in resistance arteries, in particular, depend on high energy-demanding metabolic processes sustained by CK to contract and keep their tension [23–25]. Hence, any increase in CK levels could considerably affect resistance arteries contractility and potentially blood pressure [26,27]. Here, the statistical significance of the log CK-DBP

association, despite its weak correlation coefficient, indicates that even slight variations in CK activity may influence vascular tone or indicate underlying hemodynamic stress. However, the association was not retained after multi-variable adjustment, indicating that shared variance with factors such as sex, body size, renal indices, or electrolyte status likely explains the crude signal.

A similar pattern was noted for log CK-MB: the initial correlations with SBP/DBP and PWV were apparent; however, they did not persist following adjustment.

The reduction in blood pressure variability in treated individuals may intensify cross-sectional correlations. Our findings do not support an independent association between these enzymes and arterial stiffness or blood pressure in this sample.

In a treated population, antihypertensive therapy narrows the distribution of blood pressure, attenuating cross-sectional associations with biomarkers such as CK or CK-MB. Accordingly, in this sample we find no evidence of an independent association between CK isoenzymes and arterial stiffness or blood pressure. Even so, the univariable patterns remain physiologically plausible and hypothesis-generating: in this community-based, cross-sectional cohort, log CK showed a statistically significant but small association with overall cardiovascular health, consistent with its systemic energy-metabolism role and potential to index longer-standing hemodynamic load; in men, log CK-MB displayed weak positive correlations with SBP and CVD-H, compatible with its greater cardiac specificity and putative indexing of short-term myo-cardial workload. These modest signals warrant confirmation in larger, longitudinal cohorts.

### 4.3. Electrolyte balance and cardiovascular function

Electrolytes fell within expected physiologic ranges at the group level. Nevertheless, in GLM analyses, serum calcium showed an independent positive association with log CK-MB, whereas associations of potassium and other electrolytes with CK-MB were attenuated after multivariable adjustment.These patterns are biologically plausible (given the roles of electrolytes in myocardial and vascular function) but modest in size and cross-sectional in design; they should be viewed as hypothesis-generating rather than clinically directive.

Elevated calcium levels, linked to elevated log CK-MB, can significantly affect cardiovascular health [23]. Calcium ions facilitate cardiac muscle contraction [23,24], but prolonged high levels can cause vascular tone [25,28,29], hypertension [30,31], arrhythmias [32,33], and cardiac tissue remodeling [34,35], impairing cardiac function [23,36] and exacerbate renal impairment [37,38].

Consistent with that biology, the observed calcium–CK-MB association may reflect subtle, concurrent myocardial work-load or metabolic demand, but does not establish causality between calcium status and CK-MB levels.

In clinical practice, total CK reflects skeletal muscle activity, whereas CK-MB is mostly linked to cardiac tissue; yet, contemporary assessments of myocardial injury usually rely on cardiac troponins. In community cohorts like ours, biomarker values typically reside within reference ranges; thus, any observed relationships are probably reflective of physiological variation rather than definitive disease. An assessment of CK/CK-MB specific to the disease is beyond the scope of this investigation. Longitudinal studies using repeated measurements, including troponin and ionized cal-cium, are essential to determine the reliability and independence of the calcium–CK-MB connection and its prognostic significance.

### 4.4. Implications for preventive strategies

In this community cohort, CK and CK-MB showed modest associations with CVD-H, which lost significance after multi-variable correction, indicating limited independent predictive value. While the patterns were biologically reasonable, they should be considered hypothesis-generating rather than clinically prescriptive.

The recognized sexual disparities and the independent calcium–CK-MB relationship highlight potential avenues for future, sex-specific and electrolyte-adjusted studies. Subsequent evaluations of CK/CK-MB as risk indicators must be lon-gitudinal and compared to established benchmarks, including blood pressure management and cardiac troponin testing.

 

Current preventive measures should primarily concentrate on established risk factors, including blood pressure management, lifestyle modifications, and maintenance of renal-electrolyte balance, as variations in CK or CK-MB within reference ranges are more likely because of physiological processes rather than pathological conditions.

### 4.5. Strengths and Limitations

A comment about the sample size is necessary. Out of the initial 7,000 cohort applicants, 246 satisfied the tight inclusion criteria and were selected for analysis. This reduction is attributable not to inadequate participation, but to the intentional elimination of incomplete records and those without essential demographic, biochemical, or cardiovascular information. The multi-stage screening yielded a uniform, well-defined sample while reducing extraneous variables. Although smaller than the original pool, this analytical selection demonstrated sufficient statistical power for the exploratory investigations performed.

This research is among the limited population-based studies examining CK)and CK-MB as potential cardiovascular biomarkers in a Middle Eastern cohort. The integration of biochemical, hemodynamic, and demographic parameters—such as pulse wave velocity and electrolyte indices—facilitated a comprehensive assessment of cardiovascular bioenergetics. The internal validity was enhanced through stringent inclusion criteria and multivariable modeling. The limited sample size diminishes generalizability, and the cross-sectional design obstructs causal inference. Residual confounding from unmeasured medications (e.g., diuretics), lifestyle, or metabolic factors cannot be excluded, and muscle mass was not directly assessed, which may affect sex-related disparities in CK levels. These cross-sectional associations cannot be used to draw inferences regarding treatment efficacy or failure because the study was not intended to evaluate antihypertensive drug response.

Future longitudinal studies with larger, more diverse populations and repeated biomarker assessments are necessary to enhance the understanding of the prognostic and mechanistic importance of CK and CK-MB in cardiovascular risk prediction.

## 5. Conclusion

In summary, CK and its muscle/brain isoenzyme (CK-MB) serve as potential supplemental markers of cardiovascular health, indicating essential bioenergetic and hemodynamic processes. Although their predictive capacity was constrained in our group, their significant associations with vascular and metabolic markers warrant additional investigation. When combined with demographic and anthropometric data, these biomarkers may enhance the precision of cardiovascular risk profiles. The current findings, while not endorsing therapeutic use, highlight the importance of integrating biochemical specificity with clinical context. Subsequent prospective studies with bigger and more diverse cohorts are essential to validate these findings and determine the prognostic and therapeutic relevance of CK and CK-MB within contemporary cardiovascular risk assessment models.

## Acknowledgments

We warmly thank the patients who take part in clinical research. Their kindness and trust enable medical progress. We are grateful to everyone who contributed to the MUMS Persian Cohort Study, including the participants. Their diligence and participation were critical to the success of this project. We are grateful to our colleagues and mentors for their crucial support and guidance. Finally, we thank our university for providing an outstanding environment for our studies, which allowed us to complete this pharmacy thesis.

## Author contributions

**Conceptualization:** Saeid Eslami, Zhila Taherzadeh.

**Data curation:** Mahla Izadpanah, Hamideh Safarian Bana, Elham Dehghani, Masoumeh Akbari, Fatemeh Asgharian Rezae, Fatemeh Yarmohammadi, Ashraf Alnattah.

**Formal analysis:** Mahla Izadpanah, Elahe Pourahmadi, Elham Dehghani, Masoumeh Akbari, Fatemeh Asgharian Rezae, Fatemeh Yarmohammadi, Ashraf Alnattah, Saeid Eslami, Zhila Taherzadeh.

**Funding acquisition:** Saeid Eslami, Zhila Taherzadeh.

**Investigation:** Mahla Izadpanah, Elahe Pourahmadi, Hamideh Safarian Bana, Elham Dehghani, Masoumeh Akbari, Fatemeh Asgharian Rezae, Fatemeh Yarmohammadi, Ashraf Alnattah.

**Methodology:** Elahe Pourahmadi, Saeid Eslami, Zhila Taherzadeh.

**Project administration:** Zhila Taherzadeh.

**Supervision:** Zhila Taherzadeh.

**Validation:** Elahe Pourahmadi, Masoumeh Akbari, Zhila Taherzadeh.

**Visualization:** Mahla Izadpanah, Elahe Pourahmadi, Hamideh Safarian Bana, Elham Dehghani, Masoumeh Akbari, Fatemeh Asgharian Rezae, Fatemeh Yarmohammadi, Ashraf Alnattah.

**Writing – original draft:** Mahla Izadpanah, Elahe Pourahmadi, Hamideh Safarian Bana, Elham Dehghani, Masoumeh Akbari, Fatemeh Asgharian Rezae, Fatemeh Yarmohammadi, Ashraf Alnattah, Saeid Eslami, Zhila Taherzadeh.

**Writing – review & editing:** Mahla Izadpanah, Elahe Pourahmadi, Hamideh Safarian Bana, Elham Dehghani, Masoumeh Akbari, Fatemeh Asgharian Rezae, Fatemeh Yarmohammadi, Ashraf Alnattah, Saeid Eslami, Zhila Taherzadeh.

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
