## [Decision Letter · Decision Letter 0]

18 Aug 2025

We look forward to receiving your revised manuscript.

Kind regards,

Eyüp Serhat Çalık

Academic Editor

PLOS ONE

“The authors gratefully acknowledge the financial support (Grant 941822) from the Vice-Chancellor for Research at Mashhad University of Medical Sciences, Iran. The results described in this paper were part of a Pharmacy student thesis.”

3. In the online submission form you indicate that your data is not available for proprietary reasons and have provided a contact point for accessing this data. Please note that your current contact point is a co-author on this manuscript. According to our Data Policy, the contact point must not be an author on the manuscript and must be an institutional contact, ideally not an individual. Please revise your data statement to a non-author institutional point of contact, such as a data access or ethics committee, and send this to us via return email. Please also include contact information for the third party organization, and please include the full citation of where the data can be found.

Additional Editor Comments:

I congratulate the esteemed authors for their unique approach to this important topic. Your manuscript is generally well written, but there are some issues that need to be addressed.

-The introduction and materials and methods sections are too long. You could shorten the paragraphs in the introduction that describe the biochemical metabolic functions of creatine phosphokinase or move them to the discussion section, avoiding repetition, of course.

-In the methodology section, the “Sample Collection and Laboratory Procedures” section is repeated twice; please combine these.

-In the “Sample Size Considerations” section, you state that you included 246 cases from 7,000 applications that met the study criteria. It would be helpful to add a flow chart on this topic and explain in the discussion section why you were able to obtain such a low number.

-I believe that not specifying the gender numbers in Table 1 is a major omission.

-Please add a section to the discussion section before the conclusion heading that explains the strengths and limitations of your study.

-In addition, your manuscript has been reviewed by two referees, and their comments are below. Please respond to all comments point by point and make the appropriate corrections to your manuscript.

Good luck.

Reviewers' comments:

Reviewer's Responses to Questions

**Comments to the Author**

1. Is the manuscript technically sound, and do the data support the conclusions?

Reviewer #1: No

Reviewer #2: Yes

2. Has the statistical analysis been performed appropriately and rigorously?

Reviewer #1: No

Reviewer #2: Yes

3. Have the authors made all data underlying the findings in their manuscript fully available?

Reviewer #1: No

Reviewer #2: Yes

4. Is the manuscript presented in an intelligible fashion and written in standard English?

Reviewer #1: No

Reviewer #2: Yes

Reviewer #1: the indroducation part is over reaching in it's relation between ck and ckmb and electrolyte imbalance ..

Rigorus methodology but excessive details.. Needs trimming down..

Due to the limited sample size, the focus was on descriptive analysis and assessing associations rather than hypothesis testing.

shouldn't the main study primary and secondary goals have been already established in the protocol at clinicaltrails.gov does this mean they were changed.. This point needs clarification.

individuals with low muscle mass are excluded but it doesn't say the cut off BMI for exclusion and it doesn't add the BMI in the descriptive statistics section or comment on it in the manuscript

Patients that use of captopril, colchicine, alcohol, lovastatin, lithium, lidocaine, and propranolol.

This need explanation to why patients using these specific drugs specifically were excluded and why other categories of drugs were not excluded like ccb and diuretics

( for example.. Why captopril specifically and not whole ACEI class)

Also since some of the sample size were hypertensive patients on treatment wouldn't some of the will be on diuretics that can attribute to some form of electrolytes imbalance or renal impairment .. This needs to be addressed (at least in the limitation section)

Across the whole study population, regardless of gender, a significant poor correlation (r < 0.29) was found between log CK and potassium, creatinine, systolic blood pressure (SBP), and diastolic blood pressure (DBP). A significant effect of log CK on DBP and CVH was found , meaning that each associated variable's value rose with every unit increase in log CK.

(The first sentence contradicts the second sentence regarding DBP please elaborate on this more)

The results section can be rewritten for clarity (to be less confusing)

Log CK is a significant predictor of gender.. Is it really or is it the muscle mass??

log CK levels emerged as significant predictors of CVH but Log CK-MB was found to have a significant poor correlation (r < 0.29) with both SBP and CVH in males ( please try to give an explanation or a comment on that in the discussion)

The discussion section about the hypertension and CK and CKMB is very informative

And the acknowledgement of the small sample size is on point

The manuscript lacks a didicated limitations section.

The statistics needs to clarify the numbers of males and females and make specific tables for their results and characteristics especially when part of the point of the study is the difference between them, same goes for the number of hypertensive patients.

The study didn't report on the LDL in the lipid profile in the results and the tables

The cardiovascular disease history that was mentioned repeatedly in the study and it's relation to log CK and log CKMB wasn't clearly defined in the results or the tables.

The manuscript needs correlates the results of of log ck and log ckmb to BMI and muscle mass.

There is a need in the manuscript to elaborate the cause of using the log of ck and ckmb instead of comparing them directly.

The correlation of almost all the variables is weak but the manuscript keeps making strong conclusions or at least strong recommendations from this weak correlations.

The talk about failure of antihypertensive treatments is not supported by the weak correlation.

All the results of log ckmb are within normal and only some of the results of log ck (not specified in the study how much of them) are abnormal

Which negates most of the point of the study (or at least decrease the point of it)

Almost all the multivariate results shows no correlation and the paper keeps sourcing the results of the univariate results in the results and discussion section

The manuscript talks about the relation between ck and hypertension (ck in other diseases) but doesn't talk about ckmb (in general it should expansion in talking about the relation between ck and ckmb and their relation to other diseases

After all the talk of the relation between ck and antihypertensive failure the study doesn't apply this

Conslusion and verdict

Reviewer #2: An interesting manuscript regarding cpk and ck mb values and its correlation with gender related factors and other cardiovascular risk factors. Those data are partially known nut the multivariate analysis performed by the authors confirmed this knowledge, establishing very well the correlation between these widely available laboratory parameters and cv risk factors.

**Do you want your identity to be public for this peer review?** For information about this choice, including consent withdrawal, please see our Privacy Policy

Reviewer #1: No

Reviewer #2: **Yes: ** Dimitrios Afendoulis

---

## [Author Response · Author response to Decision Letter 1]

23 Oct 2025

Dear Editor,

We sincerely appreciate your constructive feedback and the opportunity to revise and resubmit our manuscript. We have carefully considered all editorial and reviewer comments and revised the manuscript accordingly. Below, we provide a detailed point-by-point response to the editorial requirements and comments. Revised sections have also been highlighted in the revised version for your convenience.

We would like to note one minor editorial adjustment made during this revision. To increase accuracy and consistency with the study design, the publication title has been changed to indicate that the analysis is cross-sectional rather than prospective. The study's scope and content remain unchanged.

Response to Journal Requirements

Response: We have revised the entire manuscript and supporting files to align with PLOS ONE’s style guidelines and file-naming conventions, as specified in the provided template links. All headings, tables, figures, citations, and document structures now follow the required journal format.

“The authors gratefully acknowledge the financial support (Grant 941822) from the Vice-Chancellor for Research at Mashhad University of Medical Sciences, Iran. The results described in this paper were part of a Pharmacy student thesis.”

Response: As requested, we have removed funding-related text from the Acknowledgments section. The revised Funding Statement now reads as follows:

The Acknowledgments section has been retained only for non-financial contributions.

“We warmly thank the patients who participate in clinical research. Their kindness and trust enable medical progress. We are grateful to everyone who contributed to the MUMS Persian Cohort Study, including the participants. Their diligence and participation were critical to the success of this project. We are grateful to our colleagues and mentors for their crucial support and guidance. Finally, we thank our university for providing an outstanding environment for our studies, which allowed us to complete this pharmacy thesis. ”

3. In the online submission form you indicate that your data is not available for proprietary reasons and have provided a contact point for accessing this data. Please note that your current contact point is a co-author on this manuscript. According to our Data Policy, the contact point must not be an author on the manuscript and must be an institutional contact, ideally not an individual. Please revise your data statement to a non-author institutional point of contact, such as a data access or ethics committee, and send this to us via return email. Please also include contact information for the third party organization, and please include the full citation of where the data can be found.

Response: In compliance with your Data Policy, the data availability statement has been revised to indicate a non-author institutional contact point as follows:

“The datasets generated and analyzed during the current study are not publicly available due to ethical restrictions and privacy concerns regarding human participant data, as mandated by the Institutional Review Board (IRB) of Mashhad University of Medical Sciences. However, de-identified data can be made available to qualified researchers upon reasonable request, subject to approval by the MUMS IRB. Data access inquiries should be directed to the MUMS Ethics Committee at ethics@mums.ac.ir or cohort@mums.ac.ir.”

Response: The ethics statement has been retained solely within the Materials and Methods section. All duplicate mentions in other have been removed.

Response to Additional Editor Comments

1- I congratulate the esteemed authors for their unique approach to this important topic. Your manuscript is generally well written, but there are some issues that need to be addressed.

Response: We thank Editor for their supportive comments highlighting the value of our study.

2- The introduction and materials and methods sections are too long. You could shorten the paragraphs in the introduction that describe the biochemical metabolic functions of creatine phosphokinase or move them to the discussion section, avoiding repetition, of course.

Response: We have carefully shortened the Introduction by removing redundant biochemical explanations of creatine phosphokinase. Relevant mechanistic details were relocated to the Discussion section where appropriate.

The Materials and Methods section was also condensed for clarity and reduced repetition.

3- In the methodology section, the “Sample Collection and Laboratory Procedures” section is repeated twice; please combine these.

Response: The duplicated subsections have been merged into a single, streamlined section entitled “Study design and participants” to eliminate redundancy.

4- In the “Sample Size Considerations” section, you state that you included 246 cases from 7,000 applications that met the study criteria. It would be helpful to add a flow chart on this topic and explain in the discussion section why you were able to obtain such a low number.

Response: Thank you for your thoughtful and constructive feedback on our manuscript. You raise an excellent point regarding the need for greater clarity on our sample selection process. We agree that the journey from 7,000 applications to the final 246 analyzed cases is a critical aspect of our methodology that warrants a more detailed explanation. We have added a new paragraph to the Discussion section (also detailed below) that explicitly addresses the factors leading to the final sample size. We explain why this number, while a subset of the total pool, is both expected in this research context and robust for the statistical analyses performed.

“A comment about the sample size is necessary. Out of the initial 7,000 cohort applicants, 246 satisfied the tight inclusion criteria and were selected for analysis. This reduction is attributable not to inadequate participation, but to the intentional elimination of incomplete records and those without essential demographic, biochemical, or cardiovascular information. The multi-stage screening yielded a uniform, well-defined sample while reducing extraneous variables. Although smaller than the original pool, this analytical selection demonstrated sufficient statistical power for the exploratory investigations performed."

5- I believe that not specifying the gender numbers in Table 1 is a major omission.

Response: We appreciate the editor's suggestion. Because Table 1 only shows continuous variables (mean ± SD, min, max), it doesn't include categorical data like sex. But we have explicitly said how many men and women took part and what percentage of them were men and women in the Results section (3.1): "Of the 246 people who took part, 103 (41.9%) were men and 143 (58.1%) were women."

6- Please add a section to the discussion section before the conclusion heading that explains the strengths and limitations of your study.

Response: A new subsection titled Strengths and Limitations has been inserted immediately before the Conclusion. This section acknowledges study constraints (e.g., limited sample representation) and highlights the strengths, such as rigorous diagnostic criteria and standardized laboratory procedures.

Response to Reviewer Comments

Reviewer #1

1. The introduction part is over reaching in it’s relation between ck and ckmb and electrolyte imbalance .

Response: We appreciate this observation. The introduction has been revised to focus more directly on the established physiological role of CK and CK-MB in cardiovascular health. The extended mechanistic discussion of electrolyte imbalance was condensed, with only the most relevant references retained to maintain focus and balance.

2. Rigorous methodology but excessive details. Needs trimming.

Response: The methodology section has been shortened by removing redundancies (e.g., duplicate descriptions of sample collection) and summarizing technical details, while still retaining sufficient information to ensure reproducibility.

3. Due to the limited sample size, the focus was on descriptive analysis and assessing associations rather than hypothesis testing.

shouldn’t the main study primary and secondary goals have been already established in the protocol at clinicaltrails.gov does this mean they were changed.. This point needs clarification.

Response: Thank you for raising this point. The present study was designed as a descriptive and observational analysis using data derived from the MUMS Persian Cohort, rather than as a hypothesis-driven clinical trial. Accordingly, no formal sample size calculation was performed, and the study was not pre-registered in ClinicalTrials.gov, as it did not involve any interventional procedures or prospective outcome testing.

The primary objective of this work was exploratory to describe the relationships between serum CK/CK-MB levels and cardiovascular risk factors and to generate hypotheses for future confirmatory studies.

4. individuals with low muscle mass are excluded but it doesn’t say the cut off BMI for exclusion and it doesn’t add the BMI in the descriptive statistics section or comment on it in the manuscript statistics.

Response: We agree this was unclear. We have now specified that individuals with BMI < 18.5 kg/m² were excluded as proxy for low muscle mass. BMI has been added to the descriptive statistics table, and its relevance is acknowledged in the results and discussion.

5. Patients that use of captopril, colchicine, alcohol, lovastatin, lithium, lidocaine, and propranolol.

This need explanation to why patients using these specific drugs specifically were excluded and why other categories of drugs were not excluded like ccb and diuretics ( for example.. Why captopril specifically and not whole ACEI class).

•Also since some of the sample size were hypertensive patients on treatment wouldn’t some of the will be on diuretics that can attribute to some form of electrolytes imbalance or renal impairment . This needs to be addressed (at least in the limitation section)

Response: We thank the reviewer for this important comment. These specific drugs were excluded because they are known to directly alter CK/CK-MB levels (e.g., statins, colchicine, alcohol) or interfere with electrolyte balance and muscle metabolism. We agree that the rationale was not sufficiently explained. The revised manuscript now explicitly justifies these exclusions and acknowledges in the limitations section that we did not exclude all possible drug classes (e.g., calcium channel blockers, diuretics). We now discuss the possible confounding influence of diuretic therapy in hypertensive patients.

6. Across the whole study population, regardless of gender, a significant poor correlation (r < 0.29) was found between log CK and potassium, creatinine, systolic blood pressure (SBP), and diastolic blood pressure (DBP).

A significant effect of log CK on DBP and CVH was found , meaning that each associated variable’s value rose with every unit increase in log CK.

(The first sentence contradicts the second sentence regarding DBP please elaborate on this more)

Response: We thank the reviewer for this astute observation. The apparent contradiction arises from the distinction between a simple bivariate correlation and a multivariable regression model.

The poor correlation (r < 0.29) describes the isolated, linear relationship between log CK and DBP, which is indeed weak when considered alone. However, the significant effect in the regression model indicates that after accounting for the influence of other variables in the model (e.g., age, BMI, medications), log CK exerts a small but statistically independent effect on DBP. We have clarified this distinction in the revised manuscript.

7. The results section can be rewritten for clarity (to be less confusing).

Response: The results have been rewritten to improve flow and avoid confusion. Tables and text now consistently align with one another.

8. “Log CK is a significant predictor of gender—is it muscle mass instead?”.

Response: We appreciate this insightful comment. We agree that the phrasing “log CK is a significant predictor of gender is potentially misleading for a cross-sectional, observational analysis. In biomarker epidemiology, higher CK in men is widely attributed to greater skeletal muscle mass, not to sex per se as a causal factor. We have therefore

replaced “gender” with “sex (male vs. female)” throughout; reworded the Results to state that log CK differed by sex, and that this difference is likely explained by muscle mass differences; clarified in Statistical Analysis that we treat sex as a covariate, and we note BMI as a pragmatic surrogate for muscle mass in sensitivity considerations; added a statement in the Discussion acknowledging that we did not directly measure muscle mass; thus sex differences in CK should be interpreted as muscle-mass–related rather than as sex being “predicted” by CK.

9. log CK levels emerged as significant predictors of CVH but Log CK-MB was found to have a significant poor correlation (r < 0.29) with both SBP and CVH in males ( please try to give an explanation or a comment on that in the discussion).

Response: We thank the reviewer for this thoughtful point. In brief, total CK reflects systemic energy metabolism and is strongly influenced by skeletal muscle mass, whereas CK-MB is more cardiac-specific and tends to index current/subclinical myocyte stress rather than cumulative history. In our cross-sectional sample, log CK showed a small but statistically significant association with CVH, consistent with a long-term hemodynamic/ion-transport milieu, while in men the log CK-MB correlations with SBP and CVH were weak—plausibly due to CK-MB’s narrow range in non-acute settings, treatment-related SBP range restriction, smaller male subgroup size (reduced precision), and the differing biology that CK-MB reflects ongoing myocardial workload more than past events. We have added a concise clarification to the Discussion and explicitly note that these effects are modest, exploratory, and non-causal, acknowledging limitations including cross-sectional design, potential confounding by muscle mass, and the absence of direct muscle-mass measures.

10. The discussion section about the hypertension and CK and CKMB is very informative

And the acknowledgement of the small sample size is on point.

Response: We thank the reviewer for the encouraging feedback. We are pleased that the discussion of hypertension in relation to CK and CK-MB was found informative, and we appreciate the recognition of our transparent acknowledgment of the modest sample size.

Considering this comment, we made two small refinements: (i) we tightened the section on generalizability to emphasize that effect sizes should be interpreted cautiously and primarily as hypothesis-generating, and (ii) we added a brief sentence in the Limitations outlining our plan for validation in a larger, longitudinal cohort with direct measures of muscle mass. We trust these mi

---

## [Decision Letter · Decision Letter 1]

17 Nov 2025

Creatine Kinase and Its Muscle/Brain Isoenzyme as Potential Biomarkers for Cardiovascular Disease Risk: A Cross-Sectional Analysis of a Population-Based Cohort

PONE-D-25-15622R1

Dear Dr. Taherzadeh,

We’re pleased to inform you that your manuscript has been judged scientifically suitable for publication and will be formally accepted for publication once it meets all outstanding technical requirements.

Kind regards,

Eyüp Serhat Çalık

Academic Editor

PLOS ONE

Additional Editor Comments (optional):

Reviewers' comments:

Reviewer's Responses to Questions

**Comments to the Author**

Reviewer #2: All comments have been addressed

2. Is the manuscript technically sound, and do the data support the conclusions?

Reviewer #2: Yes

3. Has the statistical analysis been performed appropriately and rigorously?

Reviewer #2: Yes

4. Have the authors made all data underlying the findings in their manuscript fully available?

Reviewer #2: Yes

5. Is the manuscript presented in an intelligible fashion and written in standard English?

Reviewer #2: Yes

Reviewer #2: An interesting manuscript regarding cpk and ck mb values and its

correlation with gender related factors and other cardiovascular risk

factors. The multivariate analysis

performed by the authors confirmed this known data establishing very

well the correlation between these widely available laboratory parameters

and cv risk factors. Several factors needed to he addressed and were revised according to the suggestions of the reviewers and manuscript is fit for publication. Quite interesting work by the authors

**Do you want your identity to be public for this peer review?** For information about this choice, including consent withdrawal, please see our Privacy Policy

Reviewer #2: **Yes: ** Afendoulis Dimitrios

---

## [Editor Report · Acceptance letter]

PONE-D-25-15622R1

PLOS ONE

Dear Dr. Taherzadeh,

I'm pleased to inform you that your manuscript has been deemed suitable for publication in PLOS ONE. Congratulations! Your manuscript is now being handed over to our production team.

Kind regards,

on behalf of

Dr. Eyüp Serhat Çalık

Academic Editor

PLOS ONE